# One-Step Synthesis of LiCo_1-1.5x_Y_x_PO_4_@C Cathode Material for High-Energy Lithium-ion Batteries

**DOI:** 10.3390/ma15207325

**Published:** 2022-10-20

**Authors:** Yue Wang, Jingyi Qiu, Meng Li, Xiayu Zhu, Yuehua Wen, Bin Li

**Affiliations:** 1Research Institute of Chemical Defense, Beijing 100191, China; 2School of Materials Science and Engineering, University of Science and Technology Beijing, Beijing 100083, China

**Keywords:** lithium-ion battery, high energy, cathode, LiCo_1-1.5x_Y_x_PO_4_@C

## Abstract

Intrinsically low ion conductivity and unstable cathode electrolyte interface are two important factors affecting the performances of LiCoPO_4_ cathode material. Herein, a series of LiCo_1-1.5x_Y_x_PO_4_@C (x = 0, 0.01, 0.02, 0.03) cathode material is synthesized by a one-step method. The influence of Y substitution amount is optimized and discussed. The structure and morphology of LiCo_1-1.5x_Y_x_PO_4_@C cathode material does not lead to obvious changes with Y substitution. However, the Li/Co antisite defect is minimized and the ionic and electronic conductivities of LiCo_1-1.5x_Y_x_PO_4_@C cathode material are enhanced by Y substitution. The LiCo_0.97_Y_0.02_PO_4_@C cathode delivers a discharge capacity of 148 mAh g^−1^ at 0.1 C and 96 mAh g^−1^ at 1 C, with a capacity retention of 75% after 80 cycles at 0.1 C. Its good electrochemical performances are attributed to the following factors. (1) The uniform 5 nm carbon layer stabilizes the interface and suppresses the side reactions with the electrolyte. (2) With Y substitution, the Li/Co antisite defect is decreased and the electronic and ionic conductivity are also improved. In conclusion, our work reveals the effects of aliovalent substitution and carbon coating in LiCo_1-1.5x_Y_x_PO_4_@C electrodes to improve their electrochemical performances, and provides a method for the further development of high voltage cathode material for high-energy lithium-ion batteries.

## 1. Introduction

With the rapid development of portable devices and electric vehicles, the demand for high energy batteries is increasing. The development of high-energy cathode [1,2,3,4,5,6] and anode [7] materials is imperative. Olivine LiCoPO_4_ with a theoretical energy density of about 800 Wh kg^−1^ is a good candidate cathode material for high-energy lithium-ion batteries [8,9,10]. However, the continuous oxidative decomposition of electrolyte [11,12] and the unstable cathode electrolyte interface [13,14] under 5 V high voltage caused by Co^2+^/Co^3+^, resulting in rapid capacity degradation during cycling, severely hindering the application of LiCoPO_4_ cathode material. In addition, the Li/Co antisite exchange during the cycling process [15,16,17] and low intrinsically ionic and electronic conductivity [18,19,20] must also be overcome.

Lots of work have been undertaken to solve these problems, including decreasing the cathode particle size and controlling the morphology to shorten the Li-ion migration distance [21,22,23,24]; coating the cathode particle with stable materials [25,26,27,28] or conductive materials [29,30,31,32,33] to stabilize the interface and reduce the side reaction; partial substitution at Co site [20,34,35,36,37] to improve the intrinsic ionic and electronic conductivity [34,35,38,39,40]; and adding electrolyte additives to suppress the electrolyte decomposition [14,41,42]. Every method has some effect in improving LiCoPO_4_ cathode performance. Generally, for LiCoPO_4_ cathode material, surface coating is the most effective way to enhance the stability of the interface [25,31,43], while cation substitution can significantly improve the material ionic conductivity [16,36,44,45].

In this work, the LiCo_1-1.5x_Y_x_PO_4_@C cathode material, the substitution of Y, and carbon coating are synthesized in one step. The amount of Y substitution is optimized and its influence discussed.

## 2. Materials and Methods

### 2.1. Synthesis of LiCo_1-1.5x_Y_x_PO_4_@C Cathode Material

LiCo_1-1.5x_Y_x_PO_4_@C (x = 0, 0.01, 0.02, 0.03) cathode material was synthesized by a one-step method. First citric acid (CA, Sinopharm Chemical Reagent, ≥99.5%), Y(NO_3_)_3_·6H_2_O (Sinopharm Chemical Reagent, ≥99.0%), Co(NO_3_)_2_·6H_2_O (Sinopharm Chemical Reagent, ≥98.5%), LiNO_3_ (Sinopharm Chemical Reagent, ≥99.9%) and NH_4_H_2_PO_4_ (Sinopharm Chemical Reagent, ≥99.0%) were dissolved at stoichiometric amounts (n_Li_:n_Co_:n_Y_:n_P_:n_CA_ = 1.05:1-1.5x:x:1:2) in deionized water. Then the gel was formed by heating the solution at 80 °C. Subsequently, the wet gel was dried at 120 °C 24 h to obtain dry gel. Finally, the dry gel was calcined at 400 °C 3 h in rotary furnace at air atmosphere, and then changed to Ar atmosphere and calcined at 700 °C 2 h [39].

### 2.2. Material Characterization

The crystal information of the materials was detected by XRD (Smart Lab) with Cu Kα radiation, and TOPAS 5.0 software (Bruker AXS, America) was used to Rietveld refinements. The morphology of the materials was observed with SEM, EDS (Hitachi, BCPCAS-4800), and TEM (Tecnai, F20). FTIR spectra were obtained by IR spectrometer (PerkinElmer, Spectrum One).

### 2.3. Electrochemical Performance

The electrochemical tests were carried out with using the 2025-type coin cell. Coin cells were assembled with the dried LiCo_1-1.5x_Y_x_PO_4_@C as cathode, Li metal as anode, Celgard2400 polyethylene as separator, and 1M LiPF_6_ in a mixture of DMC/EC (*v*/*v*, 1/1) with 1 wt.% TMSB additive as electrolyte in an argon-filled glovebox. The LiCo_1-1.5x_Y_x_PO_4_@C electrodes were dried at 120 °C 12 h in a vacuum oven.

The cycling and rate performances of LiCo_1-1.5x_Y_x_PO_4_@C electrode were measured by the LAND CT2001A. The CV curve of the LiCo_1-1.5x_Y_x_PO_4_@C electrode was collected by CHI660D within the voltage range 3.0~5.3 V, with 0.05 mV s^−1^. EIS curve was conducted on Solartron SI 1260 and SI 1287 with a frequency range from 0.1 MHz to 10 MHz.

## 3. Results and Discussion

### 3.1. Composition and Morphology of LiCo_1-1.5x_Y_x_PO_4_@C Cathode Material

Figure 1 shows the XRD patterns of the four cathode materials. All four cathode materials were well indexed to the olivine structure (JCPDS: 89-6192) with an orthorhombic Pnma space group, indicating the carbon layer and Y substitution did not change the main crystal structure of LiCoPO_4_. An obvious impurity peak YPO_4_ appeared when the Y substitution amount was 0.03. This means that at the substitution amount of 0.03, it was difficult for Y to be completely incorporated into the LiCoPO_4_ lattice, as the Y atomic radius was larger than that of Co. Figure 2 presents the Rietveld refinement results of the four cathode materials. The crystal structural parameter is listed in Table 1, showing the details of the structural differences. Significantly, with the increase in the Y doping amount, the a and b parameters increased obviously, leading to an increase in unit cell volume (283.84, 284.09, 284.30 and 284.42 for x = 0, 0.01, 0.02 and 0.03). This change indicates that Y (III) replaced Co (II) and was incorporated into the LiCoPO_4_ lattice. Since the ionic radius of Y (III) (90 pm) was larger than that of Co (II) (74.5 pm), it led to an increase in the unit cell volume. The increase in the unit cell volume could facilitate the migration of Li-ion, thus improving the electrochemical performance of LiCo_1-1.5x_Y_x_PO_4_@C cathodes. According to the XRD and Rietveld refinement results, Y was successfully incorporated into the LiCoPO_4_ lattice without altering the olivine structure, although it caused an increase in the unit cell volume as Y doping amount increased.

SEM images of the four cathode materials are displayed in Figure 3a–d. All the four cathode materials had similar morphology, which was composed of agglomerated nanoparticles of about 200 nm. This suggests that the basic morphology of the four cathode materials was unaffected by Y substitution. EDS of LiCo_0.97_Y_0.02_PO_4_/C cathode material was performed to determine the Y distribution, and the result is shown in Figure 3e. It is evident that there was no region poorer or richer in Y, and the Y element was homogeneously dispersed in the LiCo_0.97_Y_0.02_PO_4_/C cathode material. The same results were observed for the O, Co, and P elements. TEM images of the four cathode materials are presented in Figure 4. All the four samples had only one diffraction fringe and had a uniform 5 nm carbon layer on the LiCoPO_4_ particle surface. From our previous studies [14,25,39], we found that a uniform carbon coating layer firstly refined particle size and improved material conductivity; secondly prevented direct contact between electrolyte and LiCoPO_4_ particle, and inhibited the continuous oxidative decomposition of electrolyte under 5 V high voltage caused by Co^2+^/Co^3+^, and thirdly, stabilized the interface between the cathode and electrolyte and suppressed the continuous generation of CEI on the LiCoPO_4_ particle surface, thus improving the electrochemical performance of the LiCoPO_4_ material. SEM and TEM images suggest that the basic morphology of the four cathode materials was unaffected by Y substitution; that is, it consisted of clustered nanoparticles with a 5 nm uniform carbon film on the Li LiCo_1-1.5x_Y_x_PO_4_@C particle surface.

Figure 5 exhibits the FTIR spectra of the four cathode materials. It is obvious that with an increase in the Y substitution, the symmetric stretching at 987 cm^−1^ position shifted to 979 cm^−1^ position when x = 0.02, which indicated the decrease in the Li/Co antisite defect [8,15,16]. However, when the Y substitution amount increased to 0.03, the symmetric stretching position shifted to 984 cm^−1^ position, meaning the Li/Co antisite defect increase decreased the electrochemical performance of the LiCo_0.955_Y_0.03_PO_4_ material.

The XPS spectrum of LiCoPO_4_/C and LiCo_0.97_Y_0.02_PO_4_/C cathode material is shown Figure 6. The spectra of the two cathode materials were similar, except for the LiCo_0.97_Y_0.02_PO_4_/C sample with the Y characteristic peak. Figure 6b shows the Co2p spectra of the LiCoPO_4_/C and LiCo_0.97_Y_0.02_PO_4_/C cathode material, which were consistent with the reported binding energy of the Co [39,46]. The Y3d spectra for LiCo_0.97_Y_0.02_PO_4_/C cathode material is displayed in Figure 6c. The Y3d spectra suggests that Y is present in the LiCo_0.97_Y_0.02_PO_4_/C cathode material and that the oxidation state of Y is +3.

### 3.2. Electrochemical Performances of LiCo_1-1.5x_Y_x_PO_4_@C Electrodes

Electrochemical performances of the four electrodes were tested in half-cell. Figure 7 shows the first cycle CV curves of the four electrodes. The CV curves of the four electrodes had two similarities: first, all had an oxidation peak around 4.3 V that can be ascribed to the electrolyte oxidation reaction; then, all had two oxidation peaks in the range of 4.8 V~4.9 V and one reduction peak around 4.7 V (the reduction potential of LiCoPO_4_ was closer and the two overlapped as a larger reduction peak at the CV test sweep rate of 0.05 mV s^−1^, which is consistent with the reports in the reference) that corresponded to the two steps of Li-ion extraction/intercalation [25,39]. The differences between the oxidation reaction potential, the reduction reaction potential, and the polarization potential of the CV curves are listed in Table 2. With increasing Y substitution amounts, the oxidation reaction potential decreased to 4.796 V and 4.892 V and the reduction potential increased to 4.71 V (x = 0.02); meanwhile, the polarization potential reduced to 0.182 V (x = 0.02). The changes in potential suggest that it is easier for Li-ion to migrate with Y substitution. These CV results reveal that the polarization of the four electrodes was reduced and that the Li-ion conductivity was enhanced with Y substitution. The improvement in Li-ion conductivity can be ascribed to the enlargement in unit cell volume of LiCo_1-1.5x_Y_x_PO_4_@C cathode material, the decrement of the Li/Co antisite defect, and the increment of the Co-site vacancy with Y substitution, which offers additional channels for Li-ion migration.

Figure 8 presents the first, second, third, tenth, twentieth, fortieth and hundredth cycle charge/discharge profiles of the four electrodes at 0.1C. The four electrodes displayed two oxidation plateaus at approximately 4.8 to 4.9 V, and two reduction plateaus at approximately 4.6 to 4.8 V, which agrees with the CV results. Noticeably, all four electrodes had one side reaction at about 4.3 V and a high overcharge capacity, which is ascribed to the electrolyte oxidation reaction. The overcharge capacity during the first charge is a major factor in the low initial coulombic efficiency [15,39]. However, at the second charge process, the overcharge capacity was reduced, which means the electrolyte oxidation reaction was restrained. This phenomenon confirms carbon film plays an important role in inhibiting the continuous oxidative decomposition of electrolyte under 5 V high voltage, and in stabilizing the interface between the cathode and electrolyte.

Figure 9 displays cycling stability and rate performances of the four electrodes. Figure 9a presents the cycling stability performance of the four electrodes at 0.1 C. The first discharge capacities of the four electrodes are 142.6, 144.5, 148, and 145.8 mAh g^−1^, respectively. After 80 cycles, the discharge capacities are 76.8, 106.6, 111, and 97.6 mAh g^−1^, with capacity retention of 53.8%, 73.7%, 75%, and 66.9%, respectively. The rapid capacity fading can be ascribed to the continuous generation of CEI on the LiCoPO_4_ particle surface and the Li/Co antisite defects during the cycling process [43,47]. Figure 9b presents the rate performances of the four electrodes. As expected, the LiCo_0.97_Y_0.02_PO_4_/C electrode displayed the best performance, with the discharge capacities of 145 mAh g^−1^ (0.1 C), 130 mAh g^−1^ (0.2 C), 113 mAh g^−1^ (0.5 C), and 96 mAh g^−1^ (1 C); the discharge capacity returned to 130 mAh g^−1^ when the discharge rate returned to 0.1 C, exhibiting good electrochemical performance stability. In comparison, the corresponding discharge capacity of the LiCoPO_4_/C electrode was about 125 mAh g^−1^ (0.1 C), 110 mAh g^−1^ (0.2 C), 82 mAh g^−1^ (0.5 C), and 66 mAh g^−1^ (1 C). The improvement of the cycling stability and rate performances for LiCo_0.97_Y_0.02_PO_4_/C electrode was ascribed to the decrease in the Li/Co antisite defect and the increase in ionic conductivity due to Y doping. Similar improvements in cycling stability and rate performance were reported for Cr-doped LiCoPO_4_ [39,48] and V-doped LiCoPO_4_ [16,37], which the authors, due to the facilitation of ion migration, caused by Cr or V substitution. Table 3 shows the cycling performance of the LiCo_0.97_Y_0.02_PO_4_/C electrode in comparison with others reported.

Figure 10 shows the EIS spectra of the four electrodes and the corresponding equivalent circuits. The simulation results are listed in Table 4. It is clear that all four cathodes had similar ohmic resistance Re (1.6 Ω, 1.33 Ω, 1.41 Ω, and 1.53 Ω, respectively) due to their similar basic forms. However, the transfer resistance Rct (59.15 Ω, 25.41 Ω, 23.66 Ω and 21.03 Ω, respectively) decreased significantly, which implies that the electronic conductivity of the LiCo_1-1.5x_Y_x_PO_4_@C cathode material was improved with the Y doping. The D_Li_^+^ results are presented in Table 4. By increasing the Y doping amounts, the D _Li_^+^ was improved, and the LiCo_0.955_Y_0.03_PO_4_@C electrode showed the best values of 6.16 × 10^−14^ cm^2^ s^−1^, whereas the LiCoPO_4_@C electrode only attained 7.11 × 10^−16^ cm^2^ s^−1^. The EIS results reveal that the intrinsic performance of ionic and electronic conductivities for LiCo_1-1.5x_Y_x_PO_4_@C material was improved by Y substitution. This improvement can be ascribed to the enlargement in unit cell volume and the increment of the Co-site vacancy caused by aliovalent Y substitution that provided a convenient pathway for Li ion migration.

## 4. Conclusions

In this work, the LiCo_1-1.5x_Y_x_PO_4_@C (x = 0, 0.01, 0.02 and 0.03) cathode material was synthesized in one step. The uniform carbon layer stabilized the interface between the cathode and electrolyte, inhibiting the continuous side reaction on the LiCoPO_4_ particle surface; meanwhile, the Y substitution decreased the antisite defect, increasing the ionic and electronic conductivities of LiCo_1-1.5x_Y_x_PO_4_@C sample. Thus, the LiCo_0.97_Y_0.02_PO_4_@C cathode exhibited the best electrochemical performance, for instance, delivering an initial discharge capacity of 148 mAh g^−1^, with a capacity retention of 75% after 80 cycles at 0.1 C, and delivered a capacity of 96 mAh g^−1^ at 1 C. The low Li/Co antisite defect, the enhancement of electronic and Li-ion conductivity caused by Y substitution, and the uniform carbon layer, worked together to improve the performance of LiCo_1-1.5x_Y_x_PO_4_@C cathode.

## Figures and Tables

**Figure 1 materials-15-07325-f001:**
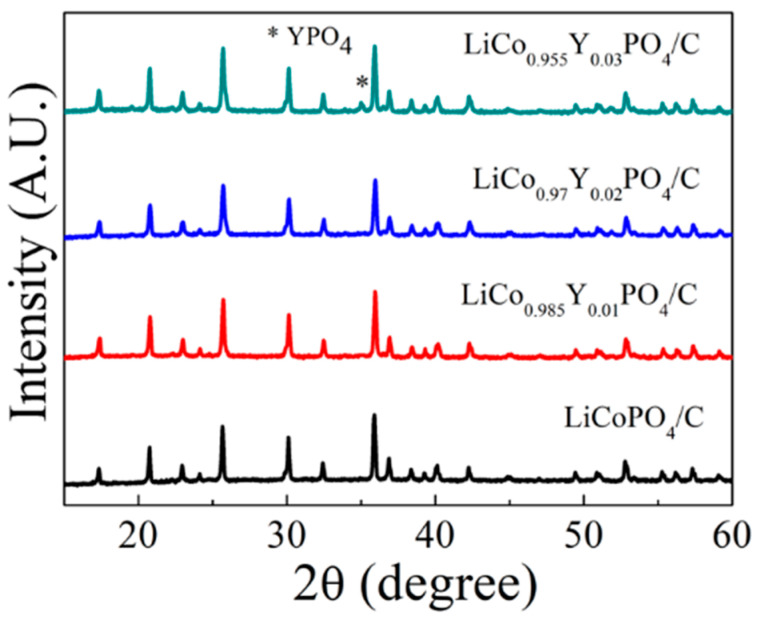
XRD patterns of the four cathode materials.

**Figure 2 materials-15-07325-f002:**
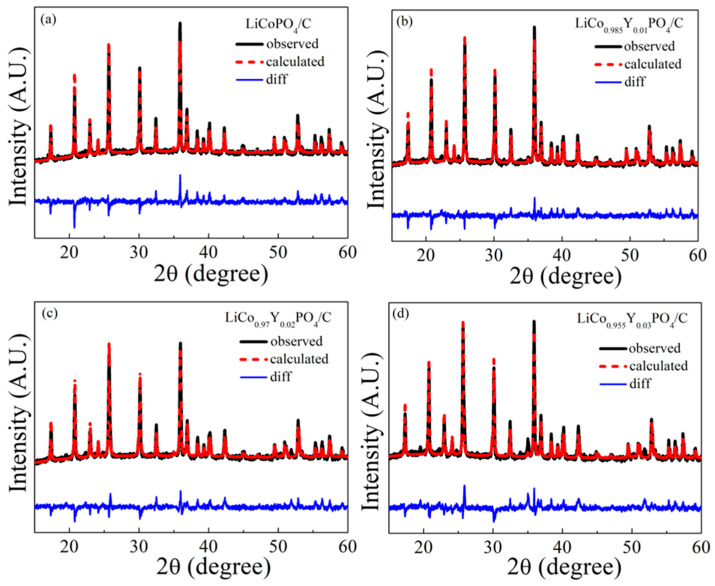
Refinement results of the four cathode materials.

**Figure 3 materials-15-07325-f003:**
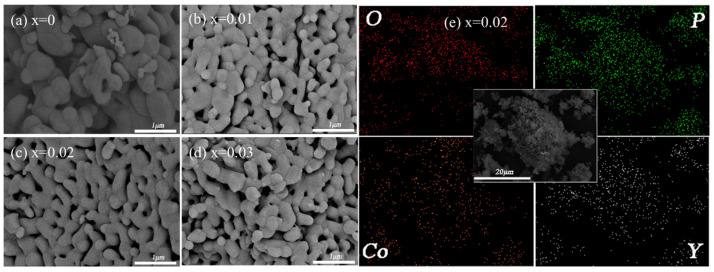
(**a**–**d**) SEM images of the four cathode materials and (**e**) EDS images of LiCo_0.97_Y_0.02_PO_4_/C cathode materials.

**Figure 4 materials-15-07325-f004:**
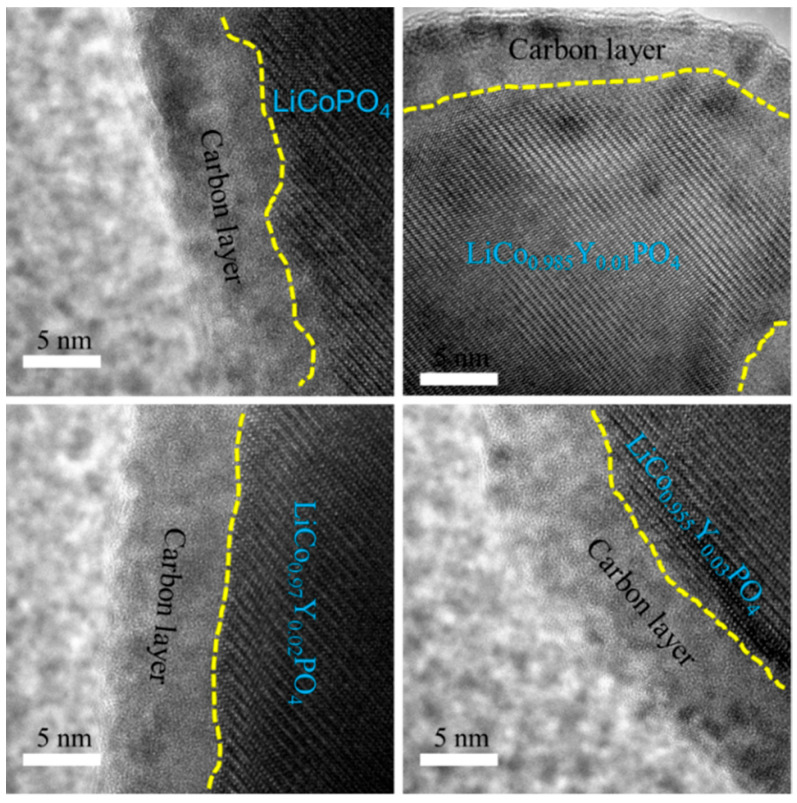
TEM images of the four cathode materials.

**Figure 5 materials-15-07325-f005:**
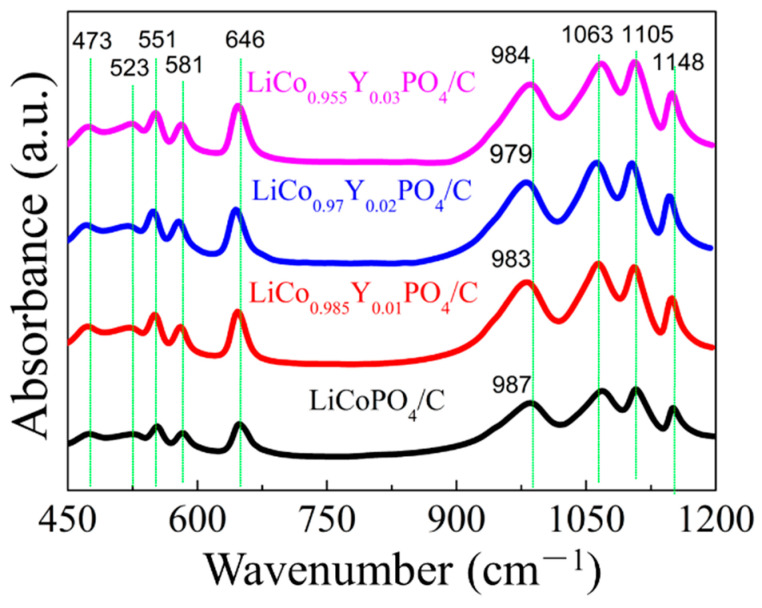
FTIR spectra of the four cathode materials.

**Figure 6 materials-15-07325-f006:**
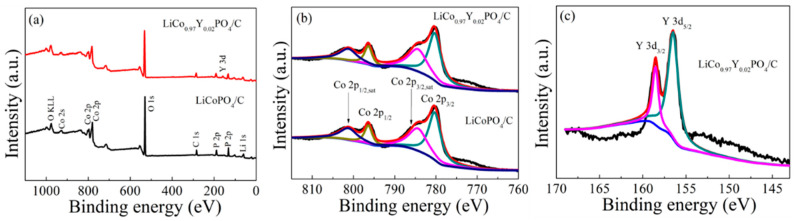
XPS spectrum of LiCoPO_4_/C and LiCo_0.97_Y_0.02_PO_4_/C cathode material.

**Figure 7 materials-15-07325-f007:**
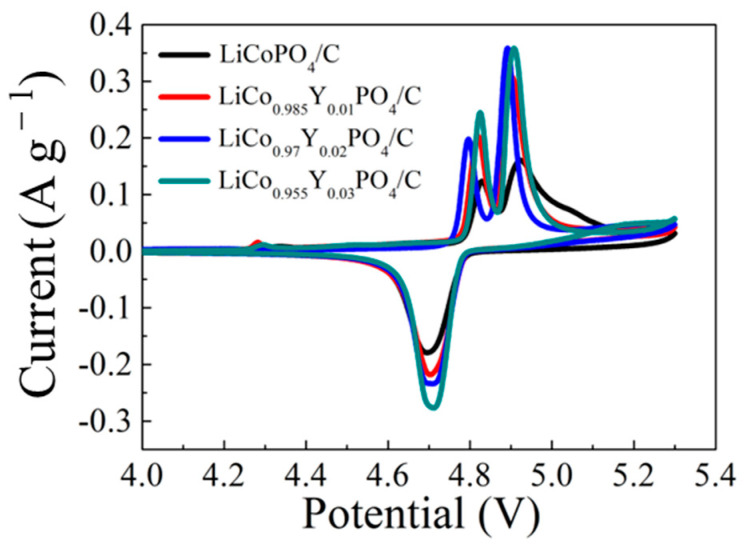
CV curves of the four electrodes.

**Figure 8 materials-15-07325-f008:**
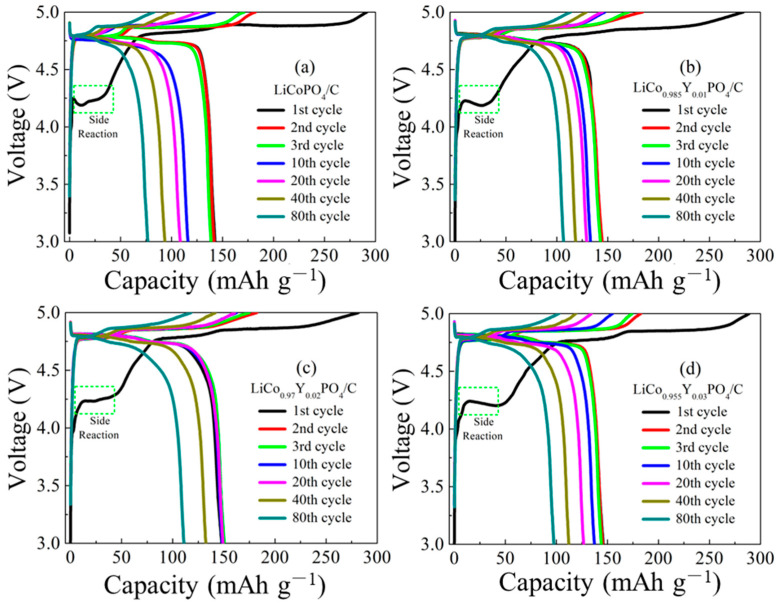
The charge/discharge profiles of the four electrodes.

**Figure 9 materials-15-07325-f009:**
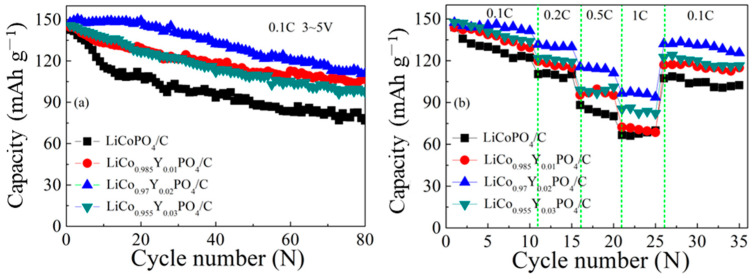
The cycling stability (**a**) and rate (**b**) performances of the four electrodes.

**Figure 10 materials-15-07325-f010:**
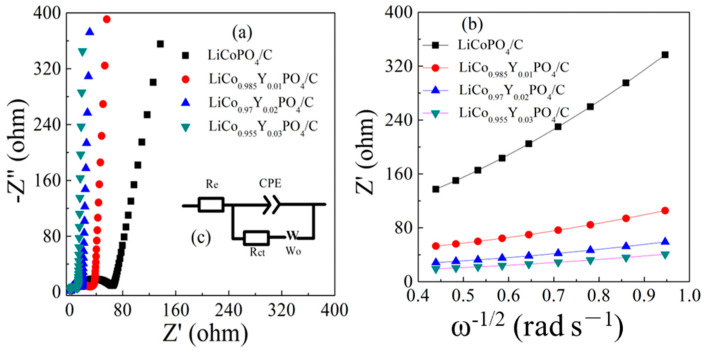
The EIS spectra of (**a**) the four electrodes and (**b**) the linear relationship between Z′ and ω^−1/2^ in four electrodes.

**Table 1 materials-15-07325-t001:** Crystal structural parameters of the four cathode materials.

Samples	a (Å)	b (Å)	c (Å)	V (Å^3^)	R_wp_	R_p_	GOF
LiCoPO_4_ (89-6192)	10.2021	5.9227	4.7003	284.01			
LiCoPO_4_@C	10.2007	5.9220	4.6987	283.84	3.85	2.71	1.73
LiCo_0.985_Y_0.01_PO_4_@C	10.2027	5.9245	4.6999	284.09	3.96	2.74	1.78
LiCo_0.97_Y_0.02_PO_4_@C	10.2058	5.9275	4.6995	284.30	4.02	2.82	1.84
LiCo_0.955_Y_0.03_PO_4_@C	10.2077	5.9288	4.6996	284.42	4.07	2.92	1.89

**Table 2 materials-15-07325-t002:** The differences of the CV curves of the four electrodes.

Samples	Oxidation Potential (V)		Reduction Potential (V)	Polarization Potential (V)
LiCoPO_4_@C	4.829	4.924	4.695	0.229
LiCo_0.985_Y_0.01_PO_4_/C	4.82	4.903	4.704	0.199
LiCo_0.97_Y_0.02_PO_4_/C	4.796	4.892	4.71	0.182
LiCo_0.955_Y_0.03_PO_4_/C	4.827	4.908	4.712	0.196

**Table 3 materials-15-07325-t003:** Cycling performance of the LiCo_0.97_Y_0.02_PO_4_/C electrode in comparison with others reported.

Samples	Rate	Initial Discharge Capacity (mAh g^−1^)	Capacity Retention (%)	Cycles	Method
Our work	0.1C	148	75	80	Y-Substituted and carbon coating
Ref. [33]	0.1C	135	52	30	Carbon coating
Ref. [49]	0.1C	147	69	40	Carbon coating
Ref. [31]	0.1C	120	75	20	Carbon coating
Ref. [32]	0.1C	124	56	100	Carbon coating
Ref. [16]	0.1C	97	85	20	V-Substituted
Ref. [37]	0.1C	145	52	20	V-Substituted
Ref. [44]	0.1C	153	21	30	Y-Substituted
Ref. [15]	0.1C	124	80	20	Fe-Substituted
Ref. [36]	0.1C	88	22	20	Mg-Substituted

**Table 4 materials-15-07325-t004:** Impedance parameters of the four electrodes.

Samples	R_e_ (Ω)	R_ct_ (Ω)	σ	*D* _Li_^+^ (cm^2^ s^−1^)
LiCoPO_4_@C	1.6	59.15	391	7.11 × 10^−16^
LiCo_0.985_Y_0.01_PO_4_/C	1.33	25.41	103	1.02 × 10^−14^
LiCo_0.97_Y_0.02_PO_4_/C	1.41	23.66	60	3.02 × 10^−14^
LiCo_0.955_Y_0.03_PO_4_/C	1.53	21.03	42	6.16 × 10^−14^

## Data Availability

Not applicable.

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
