# Peer review of "One-Step Synthesis of LiCo1-1.5xYxPO4@C Cathode Material for High-Energy Lithium-ion Batteries"

_materials, 2022, doi:10.3390/ma15207325_

Round 1

Reviewer 1 Report

The paper reports on a novel cathode material for LIB batteries. Low intrinsically ion conductivity and unstable cathode electrolyte interface are important reasons affecting the performances of Li-based cathode materials. Here a series of novel Yttrium doped cathodes are reported. The novel materials possess good electrochemical performances. The paper is well written, and the novel materials are well and thoroughly characterized.  The innovation presented in the paper does bring sufficient advances over the state of the art in my view to warrant publication in materials.  I suggest the authors cite the following relevant papers to broaden readership: DOI: 10.1021/acsami.0c22464; 10.1021/acsanm.2c02313

Author Response

Thank you for your review. Please see the attached file of author's reply.

Reviewer 2 Report

In this work, the authors have reported the synthesis of LiCo1-1.5xYxPO4@C material for LIB cathode. The manuscript could be suitable for publication in the following journal after addressing the points below correctly. I would recommend a major revision.

 1.     Authors need to address the state art of battery cathode in the first part of the introduction concerning LIB/NIB cathode, like; layered oxide cathode, polyanionic cathode, and anion redox cathode materials. It would be useful to add the following references, e.g., J. Mater. Chem. A, 2022, 10, 9941-9953; Dalton Trans., 2022, 51, 12467-12475; J. Phys. Mater. 2021, 4 024004.      

2.     How much percentage of impurity was observed in the 0.03 of Y substitution in Fig. 1?

3.    Why is one reduction peak in the CV plot (Fig. 7) as two oxidation peaks present in the anodic scan? Please describe with proper explanation in the manuscript.

4.      Is that electrolyte oxidation happening at the higher potential, ~5V? Please explain in the text.

5.     What is the side reaction shown in Fig. 8, charge/discharge plot? Please explain in the manuscript.

6.     Why LiCo0.97Y0.02PO4/C composition shows the best electrochemical performance? Please explain in the manuscript with proper experimental evidence.  

7.     Authors claim LiCo0.97Y0.02PO4/C is the best composition, but that does not satisfy the impedance spectroscopy data (Fig. 10a); please explain.

8.     Please describe the electrochemical reaction mechanism during the charge/discharge.

9. Please comment on the first cycle coulombic efficiency for all the compositions.

10. Please explain in the abstract the novelty of this work.

11. What is the importance of Y substitution in the LiCoPO4 parent lattice? It is not clear from this study; please explain appropriately in the manuscript.

Author Response

(The authors gave the same response as above.)

Reviewer 3 Report

The manuscript entitled “One-step Synthesis of LiCo1-1.5xYxPO4@C Cathode Material for High-energy Lithium-ion Batteries”. Some issues to be addressed will improve the quality of the manuscript. Therefore, I recommend this work could be published after the major revision

1.      Should the author write down the novelty of this article?

2.      The English composition requires many improvements. The authors should proofread the manuscript carefully to minimize grammatical errors.

3.      All the references mentioned in the paper should be cited in the text or vice-versa.

4.      The author, please add a comparative table for the related cathodic materials based on this study for reader's clear understanding.

5.      The author should provide a charge-discharge stability test of the lithium-ion battery  

6.      To enhance the strength of the manuscript and a broader readership range, some important references, needs to be incorporated as given below.

ACS Appl. Mater. Interfaces 2021, 13, 9, 11433–1144; Ceramics International

48, 2022, 28856-28863; J. Mater. Chem. A, 2018,6, 14483-14517

Author Response

(The authors gave the same response as above.)

Round 2

Reviewer 2 Report

The present form of the manuscript is suitable for publication.

Reviewer 3 Report

The author solved all comments carefully, I recommended accepting in the present form.